# Resilience of Colombian Entrepreneurships during COVID-19 Pandemic Crisis

Oscar Leonardo Acevedo [1,*], Manuel Méndez-Pinzón [1] and Adela Margarita Vélez-Rolón [2]

1   School of Basic Sciences, Institución Universitaria Politécnico Grancolombiano, Calle 57 No. 3-00 Este, Bogotá 110231, Colombia
2   School of Business, Colegio de Estudios Superiores de Administración, Diagonal 35 No. 5A-57, Bogotá 110311, Colombia
*   Correspondence: olacevedo@poligran.edu.co

**Abstract:** MSMEs were facing major challenges driven by uncontrollable macro-environmental factors caused by the COVID-19 pandemic. This paper shows the strategies that a sample of Colombian start-ups developed for their survival in the first months of the COVID-19 crisis and the close relationship between these strategies and the aid policies deployed by the Colombian government. This study involved 220 Colombian enterprises from different sectors of the economy (agriculture, commerce, services, and industry) and different stages of the life cycle. Our statistical analysis was based on a deep survey of highly significant rank correlation, the qualitative association between variables, a structural equation model, and mediation analysis. Among our results, we highlight a high dependence on ICT technologies, varied access to economic aid, and strong dependence on negative crisis impacts with the need for aid and deep business changes. Concerning the last conclusion, we found a significant mediating effect of the adaptability of the start-ups in order to explain why older, bigger, and more necessary enterprises were the ones that obtained aid.

**Keywords:** resilience; SMEs; Latin American entrepreneurship; COVID-19 economic crisis

## 1. Introduction

Enterprises face great challenges posed by the environment and society. These challenges are largely driven by processes such as globalization, technological changes, new forms of consumption, consumer tastes and preferences, as well as fortuitous events such as the COVID-19 pandemic. Under these scenarios, enterprises must find models that allow them to face and overcome different forms of crises and seek stability in times of volatility, uncertainty, complexity, and ambiguity.

Growing economic demand based on knowledge and innovation has put different actors at the center of development, among them entrepreneurs. These are characterized by their ability to understand the rapid changes in the market, to see opportunities based on consumer needs, to take advantage of their social capital, and their flexibility and motivation in times of high uncertainty (Alves et al. 2020; Grube and Storr 2018; Liguori and Winkler 2020). These critical events imply an imbalance for any business structure, a phenomenon studied in the literature at different times of financial, environmental, and/or political turmoil (Doern et al. 2018). Social, political, and institutional crises have been common in developing economies; however, in 2020 there was a health, social and economic crisis generated by the COVID-19 pandemic, which generated moments of uncertainty at international, regional, and local levels. Therefore, the COVID-19 pandemic opened an opportunity to understand crisis management not only in terms of the government measures that have been implemented to face the crisis, but also in terms of how organizations respond to these critical situations and thus change their strategies to face the future.

Manolova et al. (2020) rightly state that the current COVID-19 crisis is exceptionally difficult to predict and will have an enormous impact. This crisis can be seen as an example of

a "black swan", i.e., atypical, unpredictable, and high-impact events (Wind et al. 2020; Winston 2020). In this context, the literature, although growing, is still scarce regarding the impact of this crisis on entrepreneurships and small businesses, especially in developing countries (Alves et al. 2020; Boianovsky 2021; Herbane 2012; Kuckertz et al. 2020). On one hand, the crisis impact will probably be greater in ventures from emerging economies (Hudecheck et al. 2020; Salamzadeh and Dana 2020). On the other hand, research shows that the exploratory strategies of start-ups are features that allow for better crisis management (Archibugi et al. 2013). Furthermore, this capacity depends on internal organizational factors (management and innovation strategies) and a country's factors such as funds, human capital, intellectual property protection, and competitiveness (Hausman and Johnston 2014).

This study aims to increase the understanding of crisis management in two main directions: the first one is about how ventures managed the crisis during the first wave of COVID-19, and the second one is about the characteristics and strategies implemented by the ventures that allowed them to overcome this wave. Our study is based on empirical data obtained from a questionnaire answered by 220 Colombian entrepreneurs on how they cope with the pandemic crisis and what strategic changes were implemented to keep their businesses afloat. The great majority of these entrepeneurships were small and based in the city of Bogotá.

Our study makes a contribution at the empirical descriptive level by giving a snapshot of the Colombian SMEs environment during the first months of the COVID-19 crisis. Thanks to our statistical association analysis, we have managed to reduce the description of this environment to four main aspects: highly associated features of the start-ups such as maturity and size, the level of shock perceived by the entrepreneur, adaptability indicators such as the number and sort of strategic changes or adopted technologies, and data concerning which economic aids deployed by the Colombian government were effectively obtained. There are several insights into the details of this descriptive simplification. For instance, we have found unexpected associations between the gender of the entrepreneur and features such as the economic sector or the size of the start-up, which opens up further research on gender studies. Additionally, as a methodological contribution, we have been able to interpret data concerning technology adoption and strategic changes (evidence of strategic exploration) as indicators of adaptability.

This work goes beyond a descriptive analysis by proposing a causal model that establishes which of the other three factors (features, shock, or adaptability) better predicts the number of aids obtained by the start-up. By itself, this model is a theoretical contribution backed by empirical data. However, the obtention of economic aids can be regarded as an indicator of how well the entrepreneurship is coping with the crisis since aids can be very important during those events (Spigel 2017). Therefore, our conclusions from the model can be valuable for risk estimation and risk management, which have practical implications.

This article is structured as follows: In Section 2, we establish a literature background on crisis management and the COVID-19 economic context in Colombia. In Section 3, we present our methods and materials, and in Section 4 we present our results. These results are discussed in Section 5. Finally, the conclusions and limitations are presented in Section 6.

## 2. Theoretical Background

### 2.1. Crisis as a Research Phenomenon

Crises are defined as sudden and unexpected events that generate a period of uncertainty, perceived as an organizational threat, requiring immediate responses even if there is no adequate preparation (Coombs 2007; Doern et al. 2018; Pearson and Clair 1998; Ratten 2020; Williams et al. 2017). In addition, crises caused by epidemiological outbreaks are a special case characterized by high scalability and unpredictable propagation, as well as a prolonged interruption in the supply and demand of goods and services (Ivanov 2020). Consequently, the impact of crises on organizations should be understood on the basis of the moment of evolution and the magnitude of the crisis (Herbane 2012), and from the

point of view that start-ups are processes rather than events, which means that they should be understood as evolving through stages (Doern et al. 2018; Williams et al. 2017).

According to Salamzadeh and Dana (2020), the main challenges for start-ups in times of crisis are financial, human resources management, support mechanisms, market, and marketing challenges. However, technology is considered one of the aspects that can help to face the crisis, since it contributes to reducing business operating costs during the crisis period (Polas and Raju 2021). In Kuckertz et al. (2020), it was analyzed how the new companies have faced immediate consequences since the appearance of COVID-19, related to the decrease in sales, threats to liquidity due to costs, barriers to access financing mechanisms, adjustments in the organizational infrastructure, and the relationship with the supply chain. In this work, it was highlighted that the less-affected companies by the crisis were the ones that had a relevant-value proposition or were prepared for it.

In general, studies on crises raise two aspects that must be taken into account: on one hand, *Crisis Management*, understood as the ability to minimize the impact; on the other hand, *Resilience*, understood as the organization's ability to use and accumulate existing resources during and after a crisis (Doern et al. 2018; Linnenluecke 2015; Williams et al. 2017). Buchanan and Denyer (2012) propose six stages in which a crisis develops, describing them as incubation, incident, crisis management, investigation, organizational learning, and implementation of lessons learned; which allows us to not only understand the crisis as a process but also to model and understand different characteristics that may vary throughout the crisis. In this sense, crisis management raises different aspects to study: the role and leadership style of the entrepreneurs (Herbane 2012; Williams et al. 2017), local production (Cappelli and Cini 2020), consumer behavior and demographics (Kaytaz and Gul 2014), and government support and funding sources (Brown and Rocha 2020; Kuckertz et al. 2020). All these are highly relevant issues for small businesses and enterprises planning for a crisis (Herbane 2012), and this planning is even more important to understand the vulnerability to crises (Doern et al. 2018).

### 2.2. Entrepreneurial Resilience in Times of Crisis

Resilience as a broad concept studies the ability of organizations to continue operating in times of crisis and the deployment of resources before, during, and after a crisis (Williams et al. 2017). It also refers to the strategies chosen at each stage and the possible transformations that may occur in the business model (Kuckertz et al. 2020). Resilience can also be analyzed from different perspectives: from the organization's response to threats, the role of employees, organizational characteristics, the flexibility of the business model, and the supply chain (Linnenluecke 2015). One of the characteristics most associated in the literature with start-ups' resilience is innovation capacity (Archibugi et al. 2013; Etemad 2020; Frare and Beuren 2021; Kuckertz et al. 2020). Entrepreneurial resilience in times of crisis should be understood as a construction or learning process. It involves understanding the period of uncertainty caused by the crisis and the opportunities that a resilient entrepreneur can take advantage of (Kuckertz and Brändle 2021; Otrachshenko et al. 2022). Resilient organizations can use internal and external knowledge resources to cope with changing situations (Ciasullo et al. 2022). This requires a high learning capacity that allows the rapid combination of resources and the development of capabilities to convert them into opportunities so that they can respond to moments of uncertainty (Schepers et al. 2021).

In Doern et al. (2018), it was concluded that the tendency towards vulnerability or resilience in small businesses depends on the experience and mentality of the organization's leader and the deployment of resources. Thus, the study of the factors that influence in one direction or the other allows the understanding of not only the phenomenon itself, but also of the capacities that governments and enterprises must deploy to respond to any crisis. Likewise, its importance lies in the fact that it allows the interpretation of changes in the business environment (Cepel et al. 2020), which is particularly relevant in fragile emerging economies. One way to be able to affront VUCA (volatile, uncertain, complex, ambiguous) environments is to rely on all stakeholders, i.e., collaborators, customers,

suppliers, and local and external authorities (Archibugi et al. 2013; Blind et al. 2017); it should be emphasized that flexibility and rapid response capacity are also required, but these features are often considered inherent to new small businesses. Finally, resilience is also linked to *strategic maturity*, i.e., having a clear value proposition and understanding how flexible it can be; a strategically mature venture can validate whether the need it is addressing remains the same in times of crisis and understand how it changes (Alos-Simo et al. 2017).

Uncertainty must be distinguished from risk and ambiguity in terms of the degree of knowledge of the consequences of an event and the probability of its occurrence (Schulman 2021). In this sense, the crisis caused by COVID-19 should be considered a period of high uncertainty that determines the preconditions for resilience, and this brings as a response the ability to find opportunities in the crisis. This is related to the entrepreneurial orientation, i.e., the capacity for innovation (Kuckertz and Brändle 2021; Pedroni 2022; Schepers et al. 2021) and the use of technology that allows the absorption of innovation from different sources (Liu et al. 2022). This capacity for innovation in times of crisis is related to the rapid iterations in the business model that entrepreneurs can make leverage on their social capital (Björklund et al. 2020; Grube and Storr 2018), so the capacity for resilience is closely related to the leadership of the entrepreneur (Kimhi et al. 2021).

### 2.3. Colombian Business Context during COVID-19 Crisis

During the state of emergency due to the COVID-19 pandemic, several structural problems of the country became visible; for instance, the high rate of informality (Departamento Nacional de Estadística 2020), which prevented access to aid to address the economic crisis. Perhaps paradoxically, Colombia is presented as the fourth country with the highest rate of entrepreneurial activity in the world, according to the report of Bosma et al. (2020). This report also concluded that Colombia improved in aspects such as access to business financing, compared to other countries in the region. Another indication of high entrepreneurial initiative during the crisis is that, in the period from 2020 to 2021, the number of active companies in Colombia had a positive variation of 7%; this contrast with the -6% variation during the period from 2019 to 2020 (Chamber of Commerce of Bogotá 2021).

The effects of the crisis generated the implementation of actions such as partial closures caused by quarantines, total closure of several non-core companies, and the search for capital for investment in changes in their businesses (biosafety, technology, change of business model). Several ventures had to make organizational changes such as the reduction in the number of employees. Other effects of the crisis included an increase in the prices of several of their products or services as a result of the increase in production costs due to shortages of inputs and raw materials, changes in the exchange rate, compensation of losses, etc. To reduce the impact generated by the economic crisis, the Colombian government designed different aids for the business sector, totaling 19 trillion (COP), focused on small and medium-sized enterprises (SMEs) as shown in Table 1. In addition, the national government is presented as a guarantor through the National Guarantee Fund (FNG) to improve the risk profile of companies.

**Table 1.** Aids designed by the Colombian government to mitigate the impact of the COVID-19 crisis on the business sector. Source: Departamento Nacional de Planeación (2021).

| | |
|---|---|
| Financial Benefits | Credit for micro-enterprises. |
| | Relief and grace periods. |
| | Longer terms and preferential conditions in re-discount loans. |
| | Expansion of the working capital line. |
| | Refinancing of liabilities. |
| | Expansion of debt quotas and guarantees to enterprises. |
| | Strengthening of guarantees for SMEs and large enterprises. |
| | Line of guarantees for bond issuance. |
| | The line for debt funds. |
| | Guarantee for invoice financing (confirming). |
| | Payroll loans. |
| Tax benefits | Tariff reduction for imports of inputs and raw materials. |
| | VAT (value-added tax) free day. |
| | National and regional tax benefits. |
| | Reduction of trade tariffs. |
| Subsidies | Subsidized payment of social benefits. |
| | Payroll subsidy. |
| Platforms and digitalization of services | Development of platforms for access to working capital. |
| | Virtual business roundtables. |

## 3. Methods and Materials

### 3.1. Sample and Population

In this work, we analyze data gathered from a self-administered questionnaire answered by 220 entrepreneurship leaders. All these people were obtained from a database of approximately 300 entrepreneurs connected to Colegio de Estudios Superiores de Administración (CESA), either as students, alumni, or consultants/partners. The questionnaires were sent via e-mail at the end of May 2020 and were answered in June 2020. This convenient sampling needs some profiling of the entrepreneurs that answered the survey in order to better characterize the population they represent. The sample predominantly has ventures based in the city of Bogotá, for 186 (84.5%) of these entrepreneurships are based exclusively in Bogotá, 15 (6.8%) are based in Bogotá and other cities in Colombia, and only 19 (8.6%) are based outside Bogotá. Additionally, most ventures were small enterprises, since 176 (80%) have less than ten collaborators, 32 (14.5%) have between ten and fifty collaborators, and only 12 (5.5%) have more than fifty collaborators. The leader's gender was relatively balanced: with 94 females (42.7%) and 126 males (57.3%). Finally, the business sectors of the data were distributed as follows: 14 in Agriculture (6.5%), 60 in Commerce (27.3%), 127 in Services (57.7%), and 19 in Industry (8.6%).

### 3.2. Variables

Besides location information, size (as the number of collaborators), leader's gender, and business sector, respondents indicated the maturity stage of their entrepreneurship (four-point scale) and the venture's age (six-point scale). Respondents also indicated whether their business demand was negatively impacted by the COVID-19 crisis (4-point scale) and how the economic reactivation of their business will be in the next three months (3-point scale). The questionnaire also asked how many and which financial aids were obtained from the Colombian government (see Table 1) and how hard it was to obtain them (4-point scale). Finally, questions about strategic changes, adaptations, and adoption of Information and Communication Technology (ICT) were issued. More details will be presented in the results section.

### 3.3. Measurements of Association

Our statistical analysis will be restricted to studying the univariate distribution of variables and the bivariate level of association between each pair of variables, i.e, we identified which pairs of variables are significantly linked and to what extent. For any pair of variables that are either ordinal, discrete with few possible values, or dichotomous, we have chosen Kendall's $\tau_b$ rank correlation to measure and test the level of association of the pair. This is a very standard non-parametric measurement of association that has very few assumptions; though it seldom reaches its extreme values of $-1$ and 1 (Berry 2018). Despite some differences in the scale of the correlation coefficients, we obtained essentially the same conclusions by analyzing other possible measures of association such as Spearman's rank and polychoric correlations. We excluded the venture's location as a variable because it was almost a constant in our sample, and the other remaining pair of variables not suitable for $\tau_b$ are the ones involving the business sector. In this latter case, we used an exact Fisher's test on contingency tables to assess the statistical significance of the association between the variables (Agresti 2018). Our conclusions are mainly focused on results whose *p*-values are under the 1% significance level (strong correlation), though we mention occasional results under the 5% level. All calculations were made with the *R* open source software (version 4.0.4), with the *Kendall* package (version 2.2) for $\tau_b$ computations.

### 3.4. Structural Equation Model and Mediation Analysis

We searched for the variables that best explained which start-ups obtained the economic aids deployed by the Colombian government. By considering the measures of association results, we condensed the explanatory observed variables into three reflective latent factors: *features*, *shock*, and *adaptability* by means of a Structural Equations Model (SEM). In this SEM, the three latent factors were seen as the input variables, and the output variable was the number of aids obtained. In the SEM results, we found a strong indication of a mediating effect of the *adaptability* latent factor on the other two. In order to confirm this mediating effect, we performed a mediating analysis Imai et al. (2010). The SEM was obtained in the *R* language, using the *lavaan* package version 0.6.11, and the mediation analysis was performed via the *mediation* package version 4.5.0 (Sales 2016). More details are given in Section 4.

### 4. Results

#### 4.1. Descriptive Analysis

We found only three variables significantly associated (with 0.05 significance) with the business sector, and they are shown in Figure 1. These variables are the leader's gender, venture size, and difficulty in obtaining economic aids. Remarkably, so many other variables showed no major distinction among the economic sectors, including the negative impact of the crisis and expected recovery. All sectors but Commerce were predominantly masculine, and all sectors, except for industry, have a very large majority of small-size businesses, with Agriculture being the only other sector with a sizable proportion of large businesses. Access to aid was, in general, not easy, since most of the respondents (55%) found it difficult to obtain economic aid. Additionally, 37.7% found that this difficulty was normal, 5.5% found it easy, and 2.27% found it very easy. In addition, the struggle to access aid was not equally difficult for all sectors, since the Agriculture sector and, to a lesser extent, the Service sector perceived significantly more difficulties.

With respect to venture's maturity, most surveyed ventures are relatively mature, for the distribution is as follows: 4.1% are at the seed and development stage, 21.8% are at the start-up stage, 55.9% are at the growth stage, and 18.2% are at the expansion stage. The venture's age had a similar pattern: 10.9% are 0 to 3 months old, 7.3% are 3 to 6 months old, 12.3% are 6 to 12 months old, 17.3% are 1 to 2 years old, 17.7% are 2 to 5 years old, and 34.5% are more than 5 years old. In Table 2, we present the $\tau_b$ bivariate analysis of most pairs of ordinal and/or dichotomous variables of our study. We found a strong correlation among age, maturity, size, and gender. The latter means that the

bigger/older/more mature a venture is, the more likely that it is managed by a male leader; which is somewhat unexpected. On the contrary, the size-maturity-age correlation is relatively easy to understand, as it is part of a typical business cycle to grow through time. These internal correlations among size, maturity, and gender do not imply that these variables correlate in the same way to other variables. For instance, bigger, older, and more mature ventures have a strong tendency to obtain more aids and a milder tendency to have easier access to them; this association with economic aids is not present with the gender variable. Additionally, bigger and more mature ventures have a significantly gloomier appreciation of the present decrease in demand and the near-future development of the crisis; while this strong correlation is absent with gender and venture size.

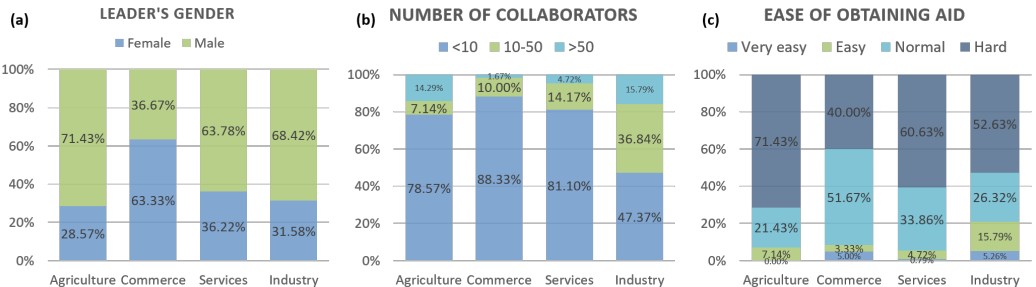

**Figure 1.** Association between venture's business sector and (**a**) leader's gender, (**b**) venture's size, or (**c**) ease of obtaining aids. Fisher's test *p*-value are, respectively: 0.0021, 0.0057, and 0.0211.

Among all the respondents, 25% state that the demand has been so slow that they have suspended operations, while 13.6% had essentially no change in demand, and 44.5% say they had a mild decrease in demand; in contrast, 16.8% of respondents claimed an increase in the demand, i.e., this crisis has been a growth opportunity and/or a new business idea for some of the ventures. About the reactivation of their business in the next three months: 29.5% answered that they expected to come back to normality very soon, 65.9% expected a gradual reactivation, while only 4.5% expected to stay closed. There is a strong and reasonable correlation between the past and expected affectations, as can be deduced from the lower-left part of Table 2. However, as in previous cases, this association does not imply that past and future affectations correlate the same with other variables. On one side, as more strategic changes are deployed, the past demand is more negatively affected, perhaps because the entrepreneur was more desperate; but this correlation is not so strong regarding the expected future, which means that these changes represent different hopes for different ventures. On the other side, the number of accessed aids does strongly correlate with both present and expected affectations.

As it is shown in Figure 2, the majority of respondents (70%) did not have access to economic aids, and ICT adoption has played a decisive role for entrepreneurs in this stage of the crisis, since only 17.7% say that they have not resorted to them. There is a strong and plausible negative correlation between the number of accessed aids and the perceived difficulty to obtain them; probably related to a lack of information or bureaucratic obstacles. Additionally, the greater the number of accessed aids, the likelier it is for the venture to have adopted technological improvements or to have deployed strategic changes in business operations. As a matter of fact, the number of accessed aids strongly correlates with all other variables of Table 2 except gender. Among the economic aids, it was established by further correlation analysis that it is payroll support that explains all these associations, while financial aid was only strongly correlated to the number of strategic changes, and all the others did not have very significant associations.

**Table 2.** *Lower left:* Kendall's rank $\tau_b$ correlation matrix of all ordinal and nominal-binary variables of the survey. *Upper right:* p-values of two-sided $\tau_b$ Kendall's test of rank correlation, where the null hypothesis value is zero correlation.

| $\tau_b$ Correlation/p-Value | Venture's Size | Venture's Age | Venture's Maturity | Leader's Gender | Number of Aids | Obstacles for Aids | Strategic Changes | ICT Embrace-ment | Present Impact | Expected Impact |
|---|---|---|---|---|---|---|---|---|---|---|
| Venture's size | | 0.000 | 0.0000 | 0.0008 | 0.0006 | 0.2657 | 0.0522 | 0.2088 | 0.0076 | 0.0080 |
| Venture's age | 0.3491 [2] | | 0.0000 | 0.0000 | 0.0000 | 0.0113 | 0.0089 | 0.5405 | 0.0001 | 0.0004 |
| Venture's maturity | 0.3455 [2] | 0.5086 [2] | | 0.0095 | 0.0007 | 0.6737 | 0.0346 | 0.5273 | 0.2774 | 0.9853 |
| Leader's Gender | 0.2218 [2] | 0.2453 [2] | 0.1655 [2] | | 0.0641 | 0.9768 | 0.1157 | 0.6354 | 0.0227 | 0.9085 |
| Number of aids | 0.2242 [2] | 0.3323 [2] | 0.2117 [2] | 0.1228 | | 0.0017 | 0.0003 | 0.0083 | 0.0028 | 0.0003 |
| Obstacles for aids | −0.0716 | −0.1483 [1] | −0.0261 | 0.0020 | −0.2012 [2] | | 0.9143 | 0.3539 | 0.2390 | 0.7889 |
| Strategic changes | 0.1250 | 0.1534 [1] | 0.1309 [1] | 0.1033 | 0.2304 [2] | −0.0069 | | 0.0590 | 0.0033 | 0.0152 |
| ICT Embracement | 0.0834 | 0.0371 | 0.0404 | 0.0322 | 0.1750 [2] | 0.0608 | 0.1239 | | 0.0988 | 0.3206 |
| Present impact | 0.1637 [2] | 0.2133 [2] | 0.0642 | 0.1425 | 0.1836 [2] | 0.0714 | 0.1781 [2] | 0.1033 | | 0.0000 |
| Expected impact | 0.1726 [2] | 0.2083 [2] | 0.0012 | 0.0077 | 0.2334 [2] | −0.0173 | 0.1564 [1] | 0.0660 | 0.4965 [2] | |

[1] Test *p*-value is below 0.05 threshold. [2] Test *p*-value is below 0.01 threshold.

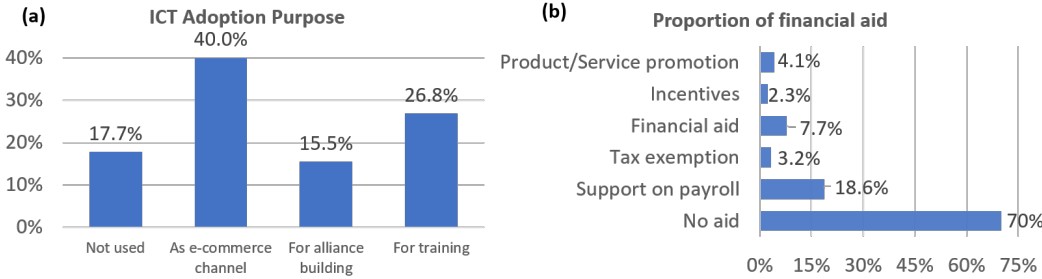

**Figure 2.** Sample distribution of (**a**) ICT adoption, and (**b**) proportion of received aids.

In Figure 3, we present the different decisions that entrepreneurs have had to make for the survival of their ventures. The strategic changes of the right part of Figure 3 are the ones that are counted in the discrete variable of Table 2. We see that this number of strategic changes is strongly correlated to the number of aids obtained and to the present negative impact, which reinforces the relationship between a negative impact and the need for more extreme measures and external assistance. We did a more detailed correlation analysis of each of these strategic changes and we found the following patterns. Change of product was not particularly associated with any other variables, which probably means that this was an all-encompassing need of different kinds of ventures. Change of commercialization channel was strongly correlated to ICT adoption since this was the most prevalent purpose, as can be seen in Figure 2a. The ventures that marked none of the strategic changes were naturally related to a low need for aid, and low past and future affectation. Similarly, a total shutdown was related to less mature ventures and more past affectation. Staff and salaries reduction behaved in a very similar manner, being strongly related to higher venture size/age/maturity and demand reduction (present impact). Finally, cancellation of orders was strongly related to the number of obtained aids, present negative impact, and higher venture age. The right part of Figure 3 shows the strategic adaptations made by the respondent entrepreneurs. These decisions are dominated by the implementation of home office, new sales channels, and new services and products. The first two decisions could be positively related to ICT adoption, but, instead, an almost insignificant negative correlation is found for new sales channels while a more significant negative correlation is found for home office. The former is easily explainable because ICT adoption has several

other purposes, so a direct correlation between these variables may well be absent; in fact, a very strong correlation is found between strategic sales channel change and ICT adoption for this very purpose. The second result is more interesting since it shows that home office is not synonymous with ICT embracement. Once more, the adoption of new services and products was not correlated significantly to any other variable.

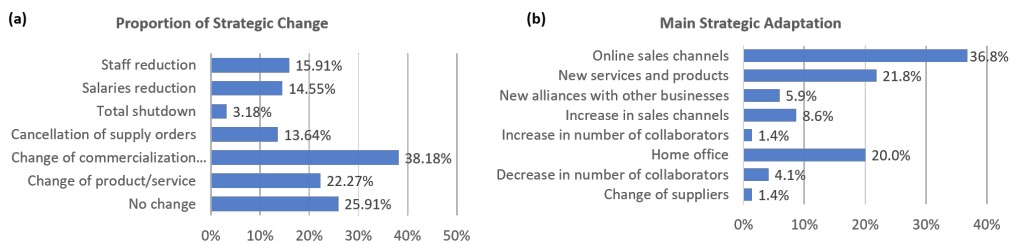

**Figure 3.** Sample proportion of (**a**) strategic changes, and (**b**) strategic adaptations.

### 4.2. Causal Analysis of Number of Aids Obtained

Besides the main descriptive analysis, where we highlighted the strongest associations between variables, we also performed a more detailed causal analysis to understand which entrepeneurships obtained the economic aids issued by the Colombian government. In Table 2, we found that several other variables were associated with the number of aids obtained; however, we also see that these variables have strong correlations between them so that we could reduce the number of explanatory factors. We performed this reduction by devising a SEM as presented in Figure 4a, where we grouped several variables in just three explanatory factors: *features*, *adaptability*, and *shock*. These factors were suggested by the association analysis of Table 2 and a theoretical reflection about the variables, while also considering the fitting measures of the SEM. The factor *features* is just attributes of start-ups (gender, number of collaborators, maturity, and age) that we found to be highly associated. The factor *shock* groups the present affectation and the future expected impact. Finally, the factor *adaptability* groups the number of strategic changes and the number of ICT adopted by the respondent managers. This last factor deserves its name because these variables are strong indicators that the enterprise can adopt several changes to cope with the crisis.

The SEM was estimated by minimizing unweighted least squares with mean and variance adjusted (ULSMV). All the observed variables were treated as ordinal; therefore, the main object of adjustment was the polychoric correlation matrix between the observed variables. The main fit indices of the SEM model are: $\chi^2$ equal to 36.451, 22 degrees of freedom, GFI of 0.998 (>0.95), AGFI of 0.993 (>0.95), RMSEA of 0.055 (<0.08), SRMR of 0.061 (<0.08), and CFI of 0.983 (>0.9). All these indexes are well inside the standard acceptance values as indicated by the reference numbers in brackets (Kline 2016; Xia and Yang 2018). The direct arrows from the three latent factors to the dependent observed variable (*aidn*) in Figure 4a have standardized weights that can vary from −1 to 1, where the extremes mean a perfect linear (or anti-linear) relation, while zero means no relation. The double-edged arrows between latent factors denote the linear correlation of each pair of them with a similar numerical interpretation.

The weights of the arrows in Figure 4a have three peculiar circumstances. First, the *shock* factor has a very light weight of 0.06 directed to the dependent variable; which contrasts with the fact that, standing alone, its associated observed variables have a good explanatory power of the dependent variable. Second, the weight of the *features* factor is appreciably lower than that of the factor *adaptability*. Finally, all three factors are relatively highly correlated. These peculiarities of the combined multivariate model are suggestive of mediation of the *adaptability* latent factor. Mediation means that the effect of the other two factors on the number of aids obtained by the enterprise is better understood as a two-step process: first from the input factor to the mediating factor, and then from the mediating factor to the output variable. In more concrete words, while it is true that the most affected

enterprises and the older ones were the ones that got the aids, it was specifically the ones among them that were more prone to adapt and make changes.

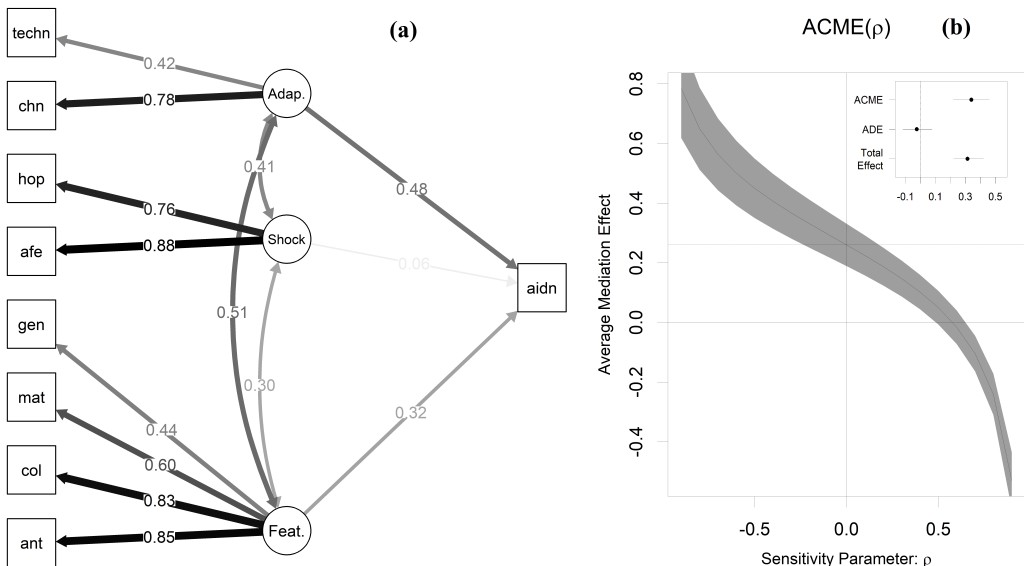

**Figure 4.** Causal analysis answering the question: which start-ups obtained the economic aids issued by the Colombian government? (**a**) SEM model that expresses the main contribution of three latent factors: start-up features, shock, and adaptability. (**b**) Mediation analysis that confirms that the predictive ability of start-up features upon having obtained aids is mediated by adaptability. *Main figure:* Sensitivity analysis figure. *Inset:* 95% confidence intervals that express how the total predictive power is splitted in mediated effect (ACME) and direct effect (ADE).

To validate our mediation conclusion, we performed a mediation analysis with the predicted values of the latent variables of the SEM and a slightly different binary dependent variable (just 0 if the enterprise did not receive aids, and 1 otherwise). This binary variable is very highly correlated to the number of aids, since only 10 out of 66 enterprises obtained more than one aid. This change of variable was necessary in order to make a mediation analysis as explained by Imai et al. (2010) using linear logistic regression, which is mathematically much less cumbersome than the equivalent analysis with an ordinal multivalued variable (Sales 2016). In the inset of Figure 4b, we have the main result of the effects of *features* on having obtained aids as mediated by *adaptability*. We found an average causal mediated effect (ACME) of 0.339, an average direct effect (ADE) of −0.026, and a total effect of 0.313. The proportion mediated is 1.082. The *p*-values of ACME and total effect (with a null hypothesis of no effect) are computationally indistinguishable from zero, which means a very strong statistical significance of our conclusion (MacKinnon and MacKinnon 2008). On the other hand, there is no good statistical significance of the direct effect ($p = 0.62$).

The mediation results are remarkable since they mean that if we suppressed the mediating effect of *adaptability*, we would find no appreciable effect of start-up *features* on having obtained aids. In other words, older, bigger, and more mature enterprises would have not obtained aids if they had not wanted to adapt. Since mediation analysis has strong assumptions that are very hard to address directly, a sensibility analysis is recommended in order to reassure the plausibility of the conclusion (Imai et al. 2010). A graph as in the main plot of Figure 4b is among the recommended sensibility analyses. The sensitivity parameter $\rho$ is the possible correlation between the error factors of the input and mediating variables. The key aspect to interpret is that the curve intercepts the horizontal axis far away from zero, which means that an unobserved variable that we have not considered in the model is unlikely to cancel the observed mediation effect (Imai et al. 2010).

## 5. Discussion

This study examined the impact that COVID-19 has had on Colombian start-ups during the first wave of the crisis (March–May 2020). The results show the differences in the levels of impact of the crisis, as well as the challenges that these types of organizations have had to face and the strategies deployed to cope with them. Colombian start-ups have managed to cope with the crisis by adopting different measures: reduction in the number of employees, change in marketing channels, cancellation of orders to suppliers, change of product or service, and reduction of salaries, up to total closure. These are the same challenges faced by start-ups in other countries (Kuckertz et al. 2020; Salamzadeh and Dana 2020).

In this research, it could be evidenced that changes in marketing channels are strongly related to the adoption of new technologies, this agrees with the literature since technology allows for reducing operating costs in a crisis according to Polas and Raju (2021). This is especially true in the stage of confinement and restriction in the first wave. However, the adoption of these new technologies will have a long-term effect (Liguori and Winkler 2020) and have been used by entrepreneurs not only to transform their marketing channels, but also for training and partnership development. It is worth mentioning that the imposition of a home office due to mandatory isolation does not necessarily result in technology adoption. In our data, it was not evident that start-ups that have explored a new product or service or new marketing channels have coped better with this period of crisis. In this sense, the start-ups' exploratory stage or innovative preparedness would imply better crisis management, i.e., it is associated with the flexibility of their business model (Archibugi et al. 2013).

On the other hand, it was also identified that start-ups strongly impacted by the crisis tend to have larger sizes and ages, but this is not related to the degree of maturity of the business. This impact may be associated with downsizing strategies and changes in the supply chain since, in general, start-ups have opted for cost reductions in order to survive. Likewise, the start-ups that made the greatest changes and those that sought to adopt technologies tended to be the larger, more mature companies. Other research evidences how companies little-affected in times of crisis possess a relevant value proposition to overcome it or because they are prepared (Williams et al. 2017).

The start-ups that did not manifest changes in their business manifested greater negative effects in the present. This is evidence that the value proposition and marketing channels are not responding to the market, and it is necessary to delve deeper into the characteristics of the entrepreneur that do not allow these adjustments to the environment (Herbane 2012; Williams et al. 2017). The misalignment between the value proposition and the market may outlive the crisis period if the necessary adjustments to the business model are not made (Lim et al. 2020). The results around ecosystem support show the relevance of government and funding aids to weather the crisis. However, they highlight the barriers these organizations have encountered to access them. Government aid was used for tax exemption, financing, payment of public service, payroll support, incentives, and promotion of their services or products through different platforms. Spigel (2017) suggests that in times of crisis, start-ups depend largely on government support and the ecosystem, so it is important to understand how these organizations have accessed these grants and how they are relying on the ecosystem to survive.

We found that the start-ups with more time in the market sought more support and felt less difficulty in obtaining it. These results can be associated with resilience to crisis, most likely developed from experience and coordination capacity (Boin and Hart 2010; Kuckertz et al. 2020; Williams et al. 2017). We also found a strong association with the degree of maturity of the start-up. Early-stage start-ups experienced greater restrictions and vulnerability to funding, as this is likely linked to sources such as close friends and family. Investment capital funding is contracted at risk since COVID-19 threatens the continuity of the business at this stage (Salamzadeh and Dana 2020). On the other hand, the sectors that expressed the greatest difficulty in accessing support are agriculture and commerce. Regarding the agrarian sector, this result is not surprising, nor is it a result

attributable solely to the pandemic. Colombia presents great lags in the implementation of public policies to strengthen the sector, a consequence of a weak institutional framework, and high levels of poverty (Departamento Nacional de Planeación 2021).

It should be noted that the Colombian ecosystem has been adapting to channel its services digitally and, at the same time, it has learned that many activities can be carried on virtually without affecting the result. It has also been identified that relationships and trust are built with social interaction, and that is why many entrepreneurs are eagerly waiting to have relationship spaces, as a complement to digital tools, which have allowed them to react quickly to the situation. Thus, both government and private actors have managed to join efforts to deliver digital knowledge for free, with tools to address the crisis, training spaces to understand government guidelines for SMEs, spaces for co-creation and development, as well as conducting rounds of capital raising.

On the other hand, there have been special spaces to support the tourism industry, spaces for health solutions, and crowdfunding initiatives for small entrepreneurs. In conclusion, the same ecosystem has seen how the solidarity evidenced in social networks, with support initiatives among individuals, has been replicated with entities and institutions that have coordinated initiatives that support small entrepreneurs and SMEs. There is a strong relationship between the characteristics of each region's ecosystem and how companies obtain resources from their environment. The importance of the relationships between different attributes demonstrates that new material attributes, such as organizations supporting entrepreneurs, state funding for start-ups, and human resource management systems, are a key factor in the success of ecosystems (Spigel 2017).

An interesting result of this research is the differences found in some aspects related to gender. We observed that, although the impact of COVID-19 does not present significant differences between male and female start-ups, the size and age of the same do. This difference is explained in the literature from different points of view. On one hand, lack of networks (Klyver and Grant 2010), barriers to access to sources of financing (Villaseca et al. 2020; Zhang et al. 2020), and the internal barrier to self-employment in women. On the other hand, entrepreneurship is deemed the new "glass ceiling", as women entrepreneurs compete in unprofitable and low-tech sectors (Villaseca et al. 2020). This conclusion coincides with the results of this study, which found greater participation of women in start-ups dedicated to commerce and trade.

Finally, we found that adaptability played a central role in explaining which start-ups obtained government financial aids, which we used as an indicator of better crisis management. For example, Doern et al. (2018) suggest two strategies for dealing with crises; the first, called crisis management, is understood as the ability to minimize the impact and companies to use and accumulate resources (such as public aid) during and after these adverse situations. The results evidence that the degree of adaptability and features of the start-ups allowed access to aid and crisis management. This coincides with other studies such as Hausman and Johnston (2014) and Cucculelli and Bettinelli (2016), which concluded that the best-adapted companies and the changes they make in times of crisis depend on internal factors of the company. Likewise, Archibugi et al. (2013) describe how companies that experience "exploratory strategies" aimed at new products and market developments are better equipped to face the crisis. Grube and Storr (2018) argue that entrepreneurs can leverage their social capital, business model flexibility, and intrinsic motivation to overcome crises; these changes associate with innovativeness and the ability to adapt to the market.

## 6. Conclusions

The results show the strategies that the start-ups in Colombia developed for their survival in the first months of the pandemic caused by COVID-19, which may not be different from the strategies implemented by start-ups in other countries. However, it highlights the close relationship between these strategies, the obtention of aid policies deployed by the government, the level of shock perceived, and the major features of

the start-ups. In particular, our results provide empirical evidence that contributes to understand the pivotal role that adaptability plays in crisis management, at least concerning which start-ups obtained aids. There is no single answer as to how the crisis should be faced and overcome, but the main practical lesson from our work is that innovative capacity is crucial for the successful revival of start-ups in the face of a crisis. More generally, our model highlights that organizational learning, maturity of the value proposition, and flexibility of the business model are key factors for success.

Despite the significant results and contributions of this research, several limitations can be found. The most egregious is the sample size, which is likely biased. Our data is limited to a short period, to a selected set of variables, and to a convenient sample of respondents. However, we consider that the data are representative and significant at the micro-level and can be interpreted in a general way. Additionally, the methodology was heavily based on statistics with limited qualitative research. Finally, our indicators of adaptability and the use of aids to assess the performance of the start-ups are rather specific; nonetheless, they provided a good approximation to the dynamics of crisis management.

Our study addresses a current and relevant issue of understanding how SMEs act in times of crisis and finding an opportunity for comparative studies with emerging economies to be expanded in future research. The results could be complemented by including micro-narratives and other qualitative data, which would consider the point of view of the entrepreneurs. Further works could address in more detail the types of strategies developed and analyze the business indicators when the process of economic reactivation ends. In addition, although the gender variable was not central to our objective, our results show strong associations that could be studied in depth. Future research could also address questions on how entrepreneurs face and build resilience in times of crisis in terms of the flexibility of their business model, prioritization of market needs, learning after the crisis, and competitive advantage; in other words, it could include other relevant variables in its study. With an expanded set of variables, it is important to carry out more multivariate analysis and to consider the effects of COVID-19 over a longer period.

**Author Contributions:** Conceptualization, A.M.V.-R. and M.M.-P.; Data curation, M.M.-P.; Formal analysis, O.L.A., A.M.V.-R. and M.M.-P.; Investigation, O.L.A., A.M.V.-R. and M.M.-P.; Methodology, O.L.A.; Writing-original draft, O.L.A., A.M.V.-R. and M.M.-P.; Writing-review and editing, O.L.A., A.M.V.-R. and M.M.-P. All authors have read and agreed to the published version of the manuscript.

**Funding:** This research received no external funding.

**Informed Consent Statement:** Participants were informed about the research and the use of the information. Informed consent was obtained from all subjects involved in the study.

**Data Availability Statement:** The data supporting this research will be made available by the authors on request and without reserve.

**Conflicts of Interest:** The authors declare no conflict of interest. The funders had no role in the design of the study; in the collection, analyses, or interpretation of data; in the writing of the manuscript; or in the decision to publish the results.

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
