# Peer review of "Resilience of Colombian Entrepreneurships during COVID-19 Pandemic Crisis"

_socsci, doi:10.3390/socsci12030130_

Round 1

Reviewer 1 Report

The authors used very relevant and contempoarry sources. The sublect is sychronous and alligned with the journal's profile. Please ckech the double fullstop "capital raising.. (432)"

I would also suggest not to write with bullets in the conclusions section.

Author Response

The authors used very relevant and contemporary sources. The subject is synchronous and aligned with the journal's profile.

Response 1: We are thankful for the positive remarks made by the referee. It is a strong encouragement for us as researchers. We also appreciate the time taken for reviewing and assessing our work.

Please check the double full stop "capital raising.. (432)"

Response 2: We have amended this typo. Thanks for the notice.

I would also suggest not to write with bullets in the conclusions section.

Response 3: We have thoroughly rewritten the conclusions section. The bullet list is no longer present. We hope that now the conclusions are better presented and more clearly related to our analysis. Besides, we have expanded the references to increase robustness and made a full revision of style, grammar, and spelling.

Reviewer 2 Report

The manuscript has quality and publication potential in Social Sciences (mdpi). This paper shows the strategies that a sample of Colombian start-ups developed for their survival in the first months of the COVID-19 crisis and the close relationship between these strategies and the aid policies deployed by the Colombian government. This study involved 220 Colombian enterprises from different sectors of the economy (agriculture, 5 commerce, services, and industry), and different stages of the life cycle. The manuscript is well developed and balanced. We suggest adding more "keywords". It will be important to increase the robustness of the manuscript (number and quality of Scopus and WoS bibliographic references). We would like further development of contributions (real, theoretical and practical) of the manuscript. The methodology section and the results / discussion show good quality and scientific consistency. We suggest that the rules / template of the manuscript be reviewed (according to the mdpi: format, referencing and template). English is competent, we suggest slight improvements through a professional / technician assessment. The study is current and very interesting. The "limitations" and "future lines of research" should be improved.

Author Response

The manuscript has quality and publication potential in Social Sciences (mdpi). This paper shows the strategies that a sample of Colombian start-ups developed for their survival in the first months of the COVID-19 crisis and the close relationship between these strategies and the aid policies deployed by the Colombian government. This study involved 220 Colombian enterprises from different sectors of the economy (agriculture, 5 commerce, services, and industry), and different stages of the life cycle. The manuscript is well-developed and balanced.

Response 1: We appreciate the detailed feedback made by the reviewer. It has helped us to greatly improve the readability and quality of our work. We hope to have made all the changes that have been suggested and to have amended all the weak points that have been pointed out.

We will proceed to explain our response to the comments and suggestions point by point.

We suggest adding more "keywords".

Response 2: We followed the suggestion by expanding the number of keywords from 3 to 4 and making them more specific.

It will be important to increase the robustness of the manuscript (number and quality of Scopus and WoS bibliographic references).

Response 3: We have added 9 references and almost all of them are from high-impact journals as desired: Björklund et al. 2020, Ciasullo et al. 2022, Kimhi et al. 2021, Kuckertz and Brändle 2021, Liu et al. 2022, Otrachshenko et al. 2022, Pedroni 2022, Schepers et al. 2021, and Schulman 2021. We present this new bibliography in the theoretical background section, adding text in lines 133 to 140 and lines 158 to 168 of the latest version.

We would like further development of contributions (real, theoretical, and practical) of the manuscript.

Response 4: We expanded our exposition of contributions in two new paragraphs in the introduction (lines 59 to 79 of the latest version) and one rewritten paragraph in the conclusions (lines 517 to 528 of the latest version). We hope that this expansion satisfactorily highlights the contributions of our work.

The methodology section and the results/discussion show good quality and scientific consistency. We suggest that the rules/template of the manuscript be reviewed (according to the mdpi: format, referencing, and template).

Response 5: Again, we appreciate the positive feedback. We used the LaTeX template of the journal and tried to comply with all the instructions for authors. We hope that the Editors will point out the parts of the manuscript that may not be properly formatted, so we can amend them.

English is competent, we suggest slight improvements through a professional / technician assessment.

Response 6: We found this is a highly pertinent recommendation. As suggested, we made a full revision of style, grammar, and spelling. Although many paragraphs have amendments, they are very slight and do not affect the meaningful content of our manuscript.

The study is current and very interesting. The "limitations" and "future lines of research" should be improved.

Response 7: We are grateful for the encouraging remark. We thoroughly rewrote the paragraphs on limitations and future lines of research in the conclusions (lines 530 to 551 of the latest version).